# Specific Gut Microbial Environment in Lard Diet-Induced Prostate Cancer Development and Progression

**DOI:** 10.3390/ijms23042214

**Published:** 2022-02-17

**Authors:** Hiromi Sato, Shintaro Narita, Masanori Ishida, Yoshiko Takahashi, Huang Mingguo, Soki Kashima, Ryohei Yamamoto, Atsushi Koizumi, Taketoshi Nara, Kazuyuki Numakura, Mitsuru Saito, Toshiaki Yoshioka, Tomonori Habuchi

**Affiliations:** 1Department of Urology, Akita University School of Medicine, Akita 010-8543, Japan; hiromisato2002@yahoo.co.jp (H.S.); 3602something@gmail.com (M.I.); yopico.t@gmail.com (Y.T.); huangmg@gipc.akita-u.ac.jp (H.M.); s4005534@gmail.com (S.K.); yama815@med.akita-u.ac.jp (R.Y.); akoizumi@med.akita-u.ac.jp (A.K.); taketonr@gipc.akita-u.ac.jp (T.N.); numakura@doc.med.akita-u.ac.jp (K.N.); urosaito@gmail.com (M.S.); thabuchi@gmail.com (T.H.); 2Field of Basic Science, Department of Occupational Therapy, Akita University Graduate School of Health Science, Akita 010-8543, Japan; yoshiokt@med.akita-u.ac.jp

**Keywords:** prostate cancer, saturated fatty acid, monounsaturated fatty acid, gut microbiota, lipid metabolism

## Abstract

Lard diet (LD) is a risk factor for prostate cancer (PCa) development and progression. Two immunocompetent mouse models fed with isocaloric specific fat diets (LD) enriched in saturated and monounsaturated fatty acid (SMFA), showed significanftly enhanced PCa progression with weight gain compared with a fish oil diet (FOD). High gut microbial divergency resulted from difference in diets, and the abundance of several bacterial species, such as in the orders Clostridiales and Lactobacillales, was markedly altered in the feces of LD- or FOD-fed mice. The proportion of the order Lactobacillales in the gut was negatively involved in SMFA-induced body weight gain and PCa progression. We found the modulation of lipid metabolism and cholesterol biosynthesis pathways with three and seven commonly up- and downregulated genes in PCa tissues, and some of them correlated with the abundance of the order Lactobacillales in mouse gut. The expression of sphingosine 1-phosphate receptor 2, which is associated with the order Lactobacillales and cancer progression in mouse models, was inversely associated with aggressive phenotype and weight gain in patients with PCa using the NCBI Gene Expression Omnibus database. Therefore, SMFA may promote PCa progression with the abundance of specific gut microbial species and overexpression of lipogenic genes in PCa. Therapeutics with alteration of gut microbiota and candidate genes involved in diet-induced PCa progression may be attractive in PCa.

## 1. Introduction

Prostate cancer (PCa) is the most commonly diagnosed cancer and the second leading cause of cancer deaths among men worldwide [1]. Epidemiological and research evidence suggest that diets and obesity have a potential to foster PCa initiation, promotion, and progression among acquired risk factors for PCa [2,3,4]. In different dietary products, saturated fatty acid (SFA) is known to increase the PCa risk, particularly advanced and fatal stages on the basis of previous population-based studies [5,6]. Moreover, accumulating evidence from in vivo studies that used a lard-based high-fat diet (HFD), which is enriched in SFA, demonstrated that SFA is more oncogenic than other dietary products [2]. In addition to the oncogenic impact of SFA, our previous study found that lipid metabolites with saturated and monounsaturated fatty acids (SMFA) were significantly higher in prostate cancer tissues than in benign prostate tissues [7]. Although several studies have yielded some findings on potential mechanisms for diet-induced PCa, including growth factor signaling, lipid metabolism, inflammation, and hormonal modulation [8,9,10], underlying mechanisms largely remain unknown.

Recent studies have reported that gut microbiota plays an important role in systemic immune response and inflammatory cytokine production [11] and is also involved in cancer development and progression [12]. With regard to the relationship between diet, obesity, and gut microbial environment, several “obese microbiota” have been identified, and specific diets influence substantial changes in gut microbiota along with the modulation of their metabolites [13,14]. In the PCa field, previous studies have evaluated gut bacterial profiles and changes after androgen deprivation therapy in patients with PCa [15,16]. A recent study showed that commensal gut microbiota contributes to hormonal resistance in castration-resistant PCa by providing an alternative source of androgens [17]. Moreover, short-chain fatty acids derived from gut microbiota of the mice fed with an HFD promote PCa growth through insulin-like growth factor (IGF) signaling [18]. The aforementioned studies support the clinical importance of interaction among diet, microbiome, and PCa development and progression. However, it is still not well defined in molecular mechanisms associated with specific diet-induced PCa carcinogenesis through gut microbiota alteration.

Although several preclinical studies have proposed that dietary fat and/or obesity enhance PCa development and progression with different underlying mechanisms, limitations have been due to the large variations in the models with the selection of preclinical models and types of dietary intervention [2]. Furthermore, many studies have adopted single preclinical models and standard diets as a control containing different ingredients [2], suggesting that more appropriate preclinical models are warranted to clarify the effect of gut microbiota profiles on specific diet-induced PCa development and progression.

In this study, we developed two immunocompetent mouse models to evaluate the role of two specific fat diets with isocaloric and an equal percentage in PCa progression. Moreover, we investigated the relationship among specific diet, gut microbiome profiles, and PCa development/progression to explore targetable bacterial species and genes associated with SMFA-enhanced PCa development and progression.

## 2. Results

### 2.1. SMFA Induces PCa Progression along with Weight Gain in Two Immunocompetent Mouse Models

In the prostate-specific Pb-Cre+ Pten^loxP/loxP^ transgenic mice (Pten KO) model, the lard diet (LD) group gained more weight (46.2 vs. 39.7 g, *p* = 0.002, Figure 1C). The mean prostate weight of the LD group was significantly higher than that of the fish oil diet (FOD) group (1463.8 vs. 678.5 mg, *p* = 0.043, Figure 1D). In the hematoxylin and eosin staining of the mouse prostate, pathological findings revealed that mouse prostate samples fed with LD showed an advanced prostatic intraepithelial neoplasia (PIN) grade compared with those fed with FOD (Figure 1E). Regarding the transgenic adenocarcinoma of the mouse prostate (TRAMP)-C2 allograft model, the mean body weight of the mice and subcutaneous tumor weight in the LD group were significantly higher than those in the FOD group (42.7 vs. 36.5 g, *p* < 0.001, 3.52 vs. 1.00 g, *p* = 0.017, respectively, Figure 1F,G). Furthermore, no difference was found in the daily calorie intake between the LD group and the FOD group in the two models (Appendix A). Collectively, these results suggested that LD significantly enhanced PCa progression with weight gain of the mice compared with FOD in the two different immunocompetent mouse models.

### 2.2. Gut Microbial Divergence in Mice Fed with LD and FOD

To assess fecal conditions in the LD and FOD groups of the two mice models, we assessed the feed efficiency, weight of dried fecal samples, and digestion efficiency in each mouse model (Figure 2A,B). The LD group had significantly higher feed efficiency than the FOD group at 10 weeks after the initiation of treatment diets in the Pten KO model (*p* = 0.009, Figure 2A). A similar tendency was observed in the TRAMP-C2 allograft model; however, this is not significant (*p* = 0.133, Figure 2B). The mean dried fecal weight in both mouse models significantly declined after switching from the control diet (CE-2) to specific fat diets regardless of the LD or FOD group (Figure 2A,B). Moreover, the mean dried fecal weight and digestion efficiency were significantly lower in the LD group than in the FOD group in both mouse models (Figure 2A,B). These results indicated that the LD facilitates digestive function in the intestine and increases feed efficiency of the mice compared with FOD.

Then, we investigated gut microbiota profiles in the LD and FOD groups of both mouse models by amplicon sequencing of the 16S ribosomal RNA gene. The alpha-diversities on the basis of the difference from the types of the models or diet were calculated by the Simpson diversity index, Shannon diversity index, and Chao1 diversity index (Figure 2C,D). When the mice were divided into two groups according to the difference of treatment diet, two of three estimators demonstrated that the gut microbiota in the LD group exhibited a significant increase in divergency than that in the FOD group (*p* = 0.019 for the Simpson diversity index, *p* = 0.023 for the Shannon diversity index, *p* = 0.128 for the Chao1 diversity index; by Mann–Whitney test, Figure 2C). By contrast, when all mice were divided into two groups based on the type of mouse models, no significant difference was noted in alpha-diversity in three different parameters for alpha divergency (*p* = 0.912 for the Simpson diversity index, *p* = 0.853 for the Shannon diversity index, *p* = 0.676 for the Chao1 diversity index; by Mann–Whitney test, Figure 2D). 

Analysis of the beta-diversity calculated on the Bray–Curtis dissimilarity and principal coordinates analysis (PCoA) revealed that the bacterial microbiota of the LD group clusters apart from that of FOD the group (permutational MANOVA, *p* = 0.005, Figure 2E). Conversely, no difference between gut microbiota in the Pten KO and TRAMP-C2 models was observed (permutational MANOVA, *p* = 0.100, Figure 2F). These data suggested that the diversity of gut microbiota profiles in the mouse model of PCa were altered by a specific fat diet condition, not a type of the mouse models. 

### 2.3. Taxonomic Differences of Gut Microbiota in the LD and FOD Groups

To identify taxonomic differences between the LD and the FOD groups, a relative taxonomic abundance was compared using linear discriminant analysis (LDA) effect size (LEfSe) analysis. The cutoff value of logarithmic LDA score of >3.0 was adopted as an important taxonomic difference between the LD and FOD groups. The taxonomic classifications revealed different gut microbiota compositions and abundances among the groups at the phyla and order levels (Figure 3A,B; Appendix A). The LEfSe identified order Clostridiales, family Lachnospiraceae, genus *Ruminococcus 2*, genus *Lachnoanaerabaculum*, genus *Marvinbryantia*, genus *Eisenbergiella*, genus *Peudobutyrivibrio*, genus *Robinsoniella*, and genus *Butyrivibrio* enriched in the LD group compared with the FOD group (Figure 3C,D; Appendix A). By contrast, the order Lactobacillales, order Bdellovibrionales, genus *Vampirovibrio*, family Bdellovibrionaceae, family Prevotellaceae, genus *Alloprebotella*, genus *Parabacteroides*, and genus *Enterorhabdus* were relatively abundant in the FOD group than in the LD group (Figure 3C,D; Appendix A). These results suggested substantial differences in gut microbiota profiles between the LD and FOD groups, and the abundance of several bacterial species, such as the orders Clostridiales and Lactobacillales, was strongly altered in the fecal microbiota between the LD and FOD groups.

### 2.4. Specific Gut Microbial Species Correlated with SMFA-Induced Weight Gain and PCa Growth

Based on the aforementioned results, we speculated that higher rates of the order Clostridiales and lower rates of the order Lactobacillales in the LD group were particularly associated with obesity and PCa development and progression. Regarding the two mouse models, the rates of the order Lactobacillales were negatively correlated with the body weight of the mice (Pten KO model; r = −0.783, *p* = 0.007, TRAMP-C2 allograft model; r = −0.802, *p* = 0.005, respectively, Figure 4A and Figure 4E). In the Pten KO model, the rate of the order Lactobacillales was negatively correlated with the mouse prostate weight (r = −0.713, *p* = 0.021, Figure 4B). Regarding the TRAMP-C2 allograft model, the order Lactobacillales had a moderate correlation with mouse subcutaneous tumor in the TRAMP-C2 allograft model (r = −0.546, *p* = 0.129, Figure 4F). On the other hand, the rate of the order Clostridiales did not correlate with weight gain and tumor volume in both models (Figure 4C,D,G,H). Collectively, the decrease in the rate of the order Lactobacillales may be involved in SMFA-induced body weight gain and PCa progression.

### 2.5. Relationship between Comprehensive Gene Expression in the Prostate and Allograft Tumors and Abundance of the Orders Lactobacillales and Clostridiales in the LD and FOD Groups

To examine underlying mechanisms of specific fat diet-induced PCa progression modulated by gut microbiota alteration, we conducted comprehensive gene expression analyses according to the differential expression levels between the LD and FOD groups using cDNA microarray analysis. In the Pten KO mouse models, 534 up- and 709 downregulated genes in the prostate of the LD group were observed, whereas 274 up- and 272 downregulated genes in the tumors of the LD group were found in the TRAMP-C2 allograft model (Appendix A). Three and seven significantly common up- and downregulated genes, respectively, in the LD group of both mouse models were identified as target genes for SMFA-enhanced PCa progression (Table 1). In 274 up- and downregulated genes in the two mouse models, the pathway enrichment analysis revealed that several pathways, including cholesterol biosynthesis (z score = 9.62), cholesterol metabolism (z score = 8.16), matrix metalloproteinases (z score = 6.72), SREBF and miR33 in cholesterol, and lipid homeostasis (z score = 6.66) were differentially regulated between the LD and FOD groups (Table 2, Appendix A). Moreover, the gene ontology (GO) analysis showed that the lipid metabolic process (z score = 5.97) was altered at the highest degree among the biological processes (Table 3, Appendix A).

Among 10 candidate genes altered between the LD and FOD groups, six genes, including *CD68m*, *SREBF2*, *TMEM45a*, *BNC1*, low-density lipoprotein receptor (*LDLR*), and sphingosine-1-phosphate receptor 2 (*S1PR2*), whose functions were previously reported [19,20,21,22,23,24], were selected for further analyses. The correlation between the mean mRNA levels of candidate genes and the rate of two bacterial species, including the orders Lactobacillales and Clostridiales, were analyzed and are shown on a heatmap (Figure 5A). The rate of the order Lactobacillales was positively correlated with the mRNA expression of three genes, including *SREBF2*, *LDLR*, and *S1PR2* (Figure 5A–D). By contrast, no significant relationship was observed between the six candidate genes and the rate of the order Clostridiales (Figure 5A). These results indicated that several genes associated with cholesterol synthesis and lipid metabolisms in cancer tissues were modulated in two mouse models with SMFA-induced PCa progression, and some of them were also correlated with the abundance of the order Lactobacillales in the mouse gut microbial environment.

### 2.6. Clinical Effect of S1PR2 and LDLR Expression in Human PCa 

Given that *S1PR2* and *LDLR* were previously reported to be associated with gut microbial conditions [25,26], the relationship between the gene expression of the two genes and the clinical outcomes of patients with PCa were explored using NCBI Gene Expression Omnibus (GEO) datasets to understand the roles of *S1PR2* and *LDLR* expressions in human PCa. We selected three datasets with several prostate tumor samples (GSE 70770, GSE21032, and GSE103512). Regarding the database of primary PCa tissue samples from patients with PCa following radical prostatectomy (*n* = 203, GSE70770), patients with a higher Gleason score (≥8) had significantly lower mRNA expressions of *S1PR2* and *LDLR* than patients with GS ≤ 7 (*p* = 0.021, *p* < 0.001, respectively, Figure 5E). In concordance with the result, another dataset of PCa tissue samples from patients with PCa (*n* = 150), including 19 patients with metastatic PCa (GSE21032), also showed that patients with GS ≥ 8 had significantly lower mRNA levels of *S1PR2* and *LDLR* in prostate tissues than those with GS ≤ 7 (*p* = 0.013, *p* < 0.001, respectively, Figure 5F). Furthermore, patients with metastasis had significantly lower *S1PR2* and *LDLR* mRNA levels in prostate tissues than those with non-metastasis (*p* = 0.002, *p* = 0.040, Figure 5G). To evaluate the effect of body mass index (BMI) on *S1PR2* and *LDLR* mRNA expression in patients with PCa, the database using paraffin-embedded tumor samples after radical prostatectomy (*n* = 50, GSE13512) showed that patients with higher BMI had a significantly lower *S1PR2* RNA levels than those with lower BMI (*p* = 0.047, Figure 5H), whereas no significant relationship was found between the *LDLR* expression and BMI (*p* = 0.289, Figure 5H). These results suggested that *S1PR2* expression in patients with PCa was inversely associated with aggressive phenotypes of PCa as well as weight gain. 

## 3. Discussion

This study shows that LD containing SMFA enhanced PCa progression along with body weight gain compared with FOD, which is rich in omega-3 polyunsaturated fatty acid (n-3 PUFA) in the two different immunocompetent mouse models. A specific fat diet condition, not the types of the mouse models, alters the gut microbiota environment, and several gut microbial species, such as the orders Lactobacillales and Clostridiales, were associated with body weight gain and PCa growth in the two mouse models. Furthermore, several target genes expressed in prostate tissues and allograft tumors, particularly related to lipid metabolism and cholesterol biosynthesis, were commonly up- and downregulated in LD mice in two mouse models and correlated with the abundance of the orders Lactobacillales and Clostridiales. Finally, *S1PR2*, which was downregulated in LD mice and correlated with abundance with the order Lactobacillales, was expressed low in patients with aggressive PCa and high BMI. There results proposed that a specific fat diet accelerated PCa development and progression through the alteration of the gut microbial environment with weight gain and target gene modulation in PCa tissues.

Accumulating evidence using in vivo models suggests that an HFD plays a role in PCa carcinogenesis [2]. However, several limitations in the preclinical models on diet-induced PCa progression have been previously highlighted [2]. Nude and severe-combined immunodeficient mice were frequently used as host mice of human PCa xenograft [8,27], indicating that the systemic condition of the hosts is very different from humans when considering the importance of the immune system in PCa progression. Differences in dietary components among research models are also known to affect the distinct effect of diet-induced metabolic disorders [28]. Several studies assessing the effect of an HFD on PCa progression have utilized a chow diet as the control treatment [29,30,31,32]. Chow is considered a high-fiber diet containing complex carbohydrates with fats from various vegetable sources and may exert significant independent unintended effects on the measured phenotypes in any research protocol [29]. Furthermore, variation of the proportion per calories of the fat component is reported to influence the tumor growth rate in the human PCa LNCaP xenografts [33]. To overcome the issues, we compared LD and FOD with isocaloric diets without difference in ingredients, except the type of fat in the two immunocompetent mouse models. In general, SFA is more oncogenic than PUFA [34], and it has been reported that monounsaturated fatty acids, such as oleic acid, may be involved in PCa progression [35]. A previous study demonstrated that the gut microbiota contributes to the metabolic phenotype in mice fed with LD or FOD through Toll-like receptor activation and white adipose tissue inflammation [36]. Consistent with previous studies, the present study enables successful confirmation about previous findings linking the effect of specific fat, gut microbial profiles, and their interaction on PCa development.

In this study, we identified the low rate of the order Lactobacillales and the high rate of the order Clostridiales observed in LD-fed mice. A recent study to compare fecal microbiota of patients with newly diagnosed treatment-naïve overweight and obese breast cancer or PCa (BMI > 25 kg/m^2^) and matched controls showed that several bacteria within the order Clostridiales were significantly different in PCa cases compared with healthy controls [37]. In particular, *Lachnospiraceae* and *Ruminococcaceae* genera were highly abundant in patients with PCa compared with cancer-free controls [37]. Recently, Pernigoni et al. developed two mouse models to assess the effect of commensal gut microbiota on castration-resistant PCa (CRPC) progression [17]. *Ruminococcus gnavus*, which is a member of the order Clostridiales, was enriched in fecal samples of castration-resistant mouse models and *Ruminococcus gnavus* administration in TRAMP-C1 mice resulted in an increasing circulating androgen level and cancer progression [17]. Similarly, Lui et al. reported that *Ruminococcus* spp. were significantly more abundant in patients with CRPC than patients with hormone-sensitive PCa [38]. Moreover, they performed fecal microbiota transplantation (FMT) in TRAMP mice and showed that FMT from feces of patients with CRPC accelerated PCa progression with a higher rate of *Ruminococcus* and increased the levels of 29 lipids, including ceramide in the feces [38,39]. In our results, *Lachnospiraceae* and *Ruminococcus 2* were also identified as markedly elevated microbial species in the LD group in the LDA analysis. Considering the results, including our data, the effect of specific species in the order Clostridiales on specific diet-induced PCa progression should be clarified, and the effect on lipid metabolism and/or androgen biosynthesis must be evaluated. By contrast, in the present study, the order Lactobacillales showed a strong negative correlation with PCa development with weight gain and was associated with several candidate genes in SMFA-enhanced PCa progression on cDNA microarrays. In accordance with the results, Liu et al. previously showed that relative abundance of the order Lactobacillales was lower (0.151%) in TRAMP mice fed with an HFD compared to those fed with a control diet (0.954%) [40]. A recent preclinical study revealed that *Lactobacillus acidophilus* attenuated obesity through gut dysbiosis with the reduction of endotoxemia and production of systematic anti-inflammatory molecules [41]. Another study reported in vitro cytotoxic effects of *L. acidophilus* LA-5 (LA-5) and *L. rhamnosus* GG (LGG) grown in the presence of oleuropein on human PCa cell lines [42,43]. Collectively, several gut microbial species of the order Lactobacillales may have a role on specific diet-induced PCa progression. Further studies are needed to elucidate the dynamism of the gut microbial environment and roles of each bacterial species on SMFA-induced PCa development and progression.

Systemic and local lipid metabolism have the potential to be modulated by gut microbiota [44]. Moreover, gut microbiota stimulates the production of unsaturated fatty acids and contributes to cholesterol production [44]. Matsushita et al. showed that short-chain fatty acids produced by gut microbiota of prostate-specific Pten KO mice fed an HFD, which mainly contains saturated fatty acids, increases blood IGF-1 and promotes PCa progression [18]. In the present study, lipid metabolism and cholesterol biosynthesis are main pathways among the comprehensive gene expression analyses using prostate tissues and tumors during SMFA-enhanced PCa development. Moreover, *LDLR* expression was significantly lower in the LD group and associated with the abundance of specific gut microbial species. Further study is warranted to determine systematic and local lipid metabolism on SMFA-induced PCa development and diet-induced dysbiosis of the gut.

*S1PR2* is a major gene of interest downregulated by LD, which is associated with the abundance of the order Lactobacillales, low BMI, and aggressive characteristics in patients with PCa. *S1PR2*, a receptor for sphingosine 1-phosphate, was reported to mediate inhibition of cell migration, invasion, and metastasis [45]. Moreover, *S1PR2* expressed in host endothelial cells and tumor-infiltrating myeloid cells in concert mediates the inhibition of tumor angiogenesis through the inhibition of vascular endothelial growth factor expression and matrix metalloproteinase 9 activity [45]. Ren et al. demonstrated that *S1PR2* downregulation promotes prostatic carcinogenesis through the stimulation of the Rac pathway and blocking of the Rho pathway [46]. Furthermore, *S1PR2* in the liver plays a key role on hepatic lipid metabolism through systematic inflammation accompanied with the gut–liver axis [30]. Further studies are required to investigate the underlying mechanism linking the relationship between *S1PR2* expression and specific diet-induced PCa progression.

This study has several limitations. First, we did not demonstrate the underlying mechanisms of gene expression changes in the prostate as well as tumor progression through SMFA-induced gut microbiota alteration. A previous study using the same genetically engineered prostate cancer model demonstrated that oral administration of an antibiotic mixture in prostate cancer–bearing mice fed with an HFD altered the composition of the gut microbiota and inhibited prostate cancer cell proliferation with the modulation of prostate IGF-1 expression [18]. The study also showed that short-chain fatty acids produced by intestinal bacteria were key regulators of HFD-induced prostate tumor growth. Moreover, microbiota-derived metabolites, including short-chain fatty acids, bile acids, and endotoxins, are known to be connected to the immune and endocrine system and associated with disease in the distant organs [47]. These studies could be partly explained by the fact that diet-induced gut microbiota alteration modulates prostate tumor growth and its gene expression through microbiota-derived metabolites. It is indispensable to clarify a more detailed mechanism underlying gene expression changes and tumor development of the prostate by SMFA-induced gut dysbiosis in future studies. Second, the preclinical models using two distinct diets used in this study is condition-specific compared with real-world dietary patterns in humans. Thirdly, this study did not assess the relationship between human fecal bacterial profiles and PCa aggressiveness. Further investigation is required to assess the interaction among detailed dietary patterns using questionnaires, including information regarding the intake of specific fat component, gut microbial patterns, and PCa aggressiveness. Finally, exploring the dynamic changes in the diet–gut microbiome on cancer development and progression is quite difficult because of its complexity. It is imperative to consider chronological changes in digested dietary products, gut microbial profiles, and activation of downstream target genes under systematic reaction after intake in the body to clarify the orchestration of thousands of microbial species and the lipidomic pattern in prostate carcinogenesis.

## 4. Materials and Methods

### 4.1. Cells

TRAMP-C2 cells [48], which were established from a prostate tumor of a TRAMP mouse, were obtained from the American Type Cell Culture Collection (Manassas, VA, USA). The cells were cultured in Dulbecco’s modified eagle’s medium (DMEM) (Invitrogen, Carlsbad, MA, USA) supplemented with 5% fetal bovine serum and 1% penicillin–streptomycin. All cells were used to establish an allograft tumor model of mice. For all in vitro experiments, the cells were subjected to no more than 15 passages.

### 4.2. Animals

We used two different immunocompetent mouse models: prostate-specific Pb-Cre+ Pten^loxP/loxP^ transgenic mice (Pten KO model) and C57BL/6 mice subcutaneously inoculated with TRAMP-C2 PCa cells (TRAMP-C2 allograft model). For the Pten KO model, PB-Cre4 mice [49] and PTEN loxP/loxP mice [50] have been described previously. PbCre4 mice and PTEN loxP/loxP mice were interbred to generate “WT” (PTEN loxP/loxP/PbCre−/−) and “PTEN-KO” (PTEN loxP/loxP/PbCre+/−) mice and backcrossed to the C57BL/6J strain for at least four generations; these mice were housed in the Akita University Animal House. All male mice used in the experiment were genotyped to identify the expression of [Cre] with [fl/fl] using the primers listed in Appendix A.

Regarding the TRAMP-C2 allograft models, male C57BL/6J mice were housed in a separate cage in a pathogen-free environment and fed an autoclaved CE-2 diet (CLEA Japan, Inc., Tokyo, Japan) until experimental diets were started. Body weight and food intake were weekly measured throughout the experiment. The institutional review board of the Akita University Graduate School of Medicine approved all animal experiments.

### 4.3. Diets

The mice were fed with two isocaloric diets comprised of different fat compositions: LD (D10011202, Research Diets, Inc., New Brunswick, NJ, USA) and FOD (D05122102, Research Diets, Inc., New Brunswick, NJ, USA). The LD contains 39.5% of lard oil, which is rich in SMFA, whereas the FOD included 39.5% of fish oil, which is rich in n-3 PUFA. Other dietary ingredients are equivalent (Appendix A). Diets were prepared and sterilized by Research Diets, Inc. The LD was stored at room temperature, and the FOD was stored in a cold room (4 °C). Feeding receptacles were on top of the cages to control food intake, and new food was given twice a week.

### 4.4. Design of Pten KO Mice Experiments

The design of the Pten KO mice experiments is described in Figure 1A. The mice (*n* = 15 each) at 11 weeks of age were randomly divided into two experimental diet groups, including the LD and FOD, and sacrificed at 28 weeks of age. We continued to feed each diet until the time of sacrifice. The mouse prostates were extracted and stored at −80 °C until further use or processed for histopathology. A half of the prostate was divided into anterior prostate and dorsal–lateral prostate/ventral prostate and separately stored at −80 °C until further use for gene expression analysis. Mouse fecal samples were collected at 21 weeks. We excluded one mouse fed with an FOD due to the presence of a non-experiment-related illness.

### 4.5. Design of TRAMP-C2 Allograft Mice Experiments

The design of TRAMP-C2 allograft mice experiments is described in Figure 1B. The mice (*n* = 15 each) were randomly divided into two different dietary groups, including the LD and FOD at 6 weeks of age. TRAMP-C2 cells (3 × 10^6^ cells suspended in 0.25 mL of DMEM) were subcutaneously injected into the hind limb of the mice at 20 weeks of age. Eight weeks after the injection, the mice were sacrificed, and the subcutaneous tumors were extracted and stored at −80 °C until further use. Mouse fecal samples were collected at 25 weeks. The tumor volume was calculated using the following formula: length (cm) × weight (cm) × height (cm) × 0.5236. One mouse fed with an FOD had a non-experiment-related illness and was excluded from the study.

### 4.6. Fecal Collection, 16S Ribosomal RNA Sequencing, and Data Processing

To collect dried fecal samples, the mice were transferred to a metabolic gauge for 2 days. Fecal samples were collected into sterilized collection tubes, immediately flash-frozen on liquid nitrogen, and stored at –80 °C until analysis. We entrusted DNA extraction from mouse fecal samples and 16S rDNA amplicon sequence analyses to TechnoSuruga Laboratory Co., Ltd. (Shizuoka, Japan), based on a previously reported method [51]. Bacterial identification from the sequence was performed according to the DB-BA 13.0 microbial identification database (TechnoSuruga Laboratory Co., Ltd., Shizuoka, Japan) with 97% similarity cutoff.

Alpha-diversity in each group was calculated using the Simpson, Shannon, and Chao1 indices, and differences of alpha-diversity among the groups were statistically tested using the Mann–Whitney test. Beta diversity was determined on the basis of the Bray–Curtis index distance method, and PCoA plots were made to explore the dissimilarity of bacterial communities between the groups. Permutational multivariate analysis of variance (PERMANOVA) was used to analyze differences in beta-diversity. Alpha- and beta-diversity analyses and visualization of the results were performed using the online software MicrobiomeAnalyst (https://www.microbiomeanalyst.ca/, accessed on 1 February 2022). The differential abundances of bacterial species between groups were identified using LEfSes analysis, and species with LDA >3.0-fold were considered significantly different. The LEfSe analysis and visualization of taxonomic cladogram were performed according to the online Galaxy-based software LEfSe (https://huttenhower.sph.harvard.edu/galaxy/, accessed on 1 February 2022).

### 4.7. RNA Extractions and Microarray Analysis

Prostates of the Pten KO mice (*n* = 6) and tumors of the TRAMP-C2 allografts (*n* = 6) were used for comprehensive mRNA expression analysis. We entrusted the RNA extraction from tissues and microarray analyses to Filgen, Inc. (Aichi, Japan). The mRNA expression profiles were determined using a GeneChip Mouse Gene 2.0 ST Array (Filgen, Inc., Aichi, Japan). Differentially expressed genes were defined as genes that showed at least a 1.25-fold change and *p*-value < 0.05 between the LD and FOD. Using the 1.25-fold change cutoff, the 274 genes that had a 1.25-fold or more difference between the LD and FOD in two mouse models were imported into the pathway analysis and GO analysis. Data were analyzed using a software package provided by Filgen, Inc. (Microarray Data Analysis Tool ver. 3.2, Aichi, Japan). The pathway analysis was conducted according to the GO database (http://geneontology.org/, accessed on 1 February 2022).

### 4.8. Bioinformatic Analysis in Human PCa Tissues

We applied the NCBI GEO, which is a database repository of high-throughput gene expression data from microarrays, to assess the effect of candidate genes in human PCa tissues. The gene expression profiles of GSE 70770, 21032, and 103512 were downloaded from the GEO database (https://www.ncbi.nlm.nih.gov/geo/, accessed on 1 February 2022). The GSE 70770, 21032, and 103512 datasets included data from 203 primary PCa tissue samples after radical prostatectomy, PCa tissue from 131 patients with non-metastatic and 19 metastatic PCa, and 50 paraffin-embedded PCa samples within 280 various cancer samples.

### 4.9. Statical Analysis

Statistical analyses were performed using SPSS vr.26 (SPSS, Chicago, IL, USA). Unpaired Student’s t-test and the Mann–Whitney test were used to compare the difference between the groups. The correlation between the two factors was tested by Spearman’s rank correlation coefficient. Difference was considered significant if *p* values were <0.05.

## 5. Conclusions

SMFA may promote PCa progression with the abundance of specific gut microbial profiles and overexpression of several lipogenic genes. Targeting gut microbiota and its downstream target genes involved in specific diet-induced PCa progression is an attractive option for novel therapeutics of PCa.

## Figures and Tables

**Figure 1 ijms-23-02214-f001:**
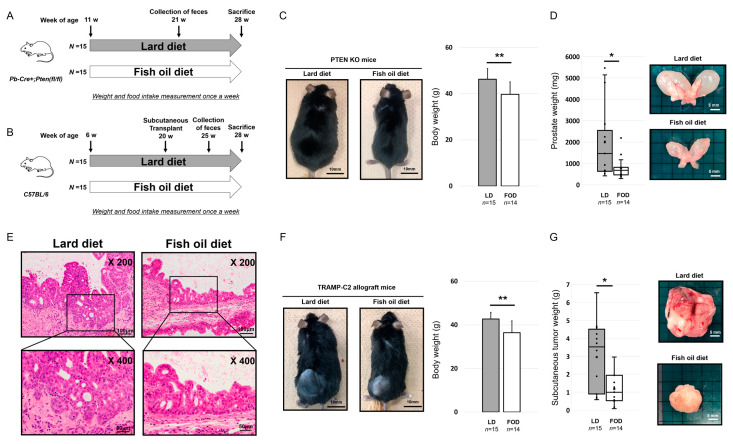
Saturated and monounsaturated fatty acids induce prostate cancer progression along with weight gain in two different immunocompetent mouse models. * *p* < 0.05, ** *p* < 0.01. (**A**) Experimental scheme of the Pten KO model. (**B**) Experimental scheme of the TRAMP-C2 allograft model. (**C**) Gross appearance and mean body weight of the Pten KO model fed with LD or FOD (*n* = 15, 14, respectively). Black bars indicate 10 mm. (**D**) Mean prostate weight and gross appearance of the prostate in the Pten KO model fed with LD or FOD (*n* = 15, 14, respectively). The line within a box indicates the median value. The round dots and cross marks indicate the value of each sample and the mean value, respectively. White bars indicate 5 mm. (**E**) Representative images of hematoxylin and eosin staining of the prostatic tissues in the Pten KO mice fed with LD or FOD. Magnification 200× in the upper images and 400× in the lower images. The scale bars indicate 100 μm in the upper images and 50 μm in the lower images. (**F**) Gross appearance and mean body weight of the TRAMP-C2 allograft model fed with LD or FOD (*n* = 15, 14, respectively). Black bars indicate 10 mm. (**G**) Mean tumor volume and gross appearances of subcutaneous tumor in the TRAMP-C2 allograft model fed with LD or FOD (*n* = 15, 14, respectively). The line within a box indicates the median value. The round dots and cross marks indicate the value of each sample and the mean value, respectively. White bars indicate 5 mm. LD, lard diet; FOD, fish oil diet.

**Figure 2 ijms-23-02214-f002:**
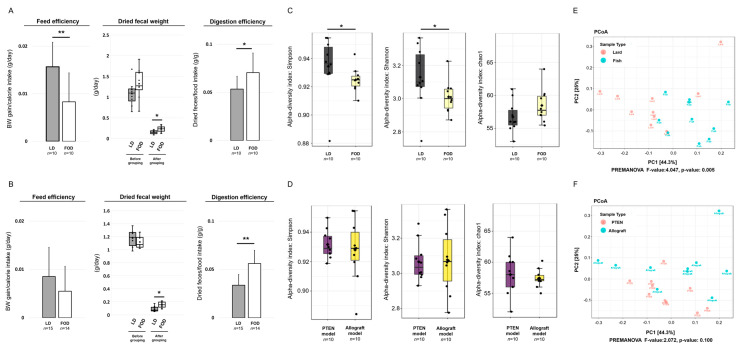
Digestive efficacy, fecal characteristics, and divergency of gut microbiota in two mouse models. * *p* < 0.05, ** *p* < 0.01. (**A**) Feed efficiency, dried fecal weight, and digestion efficiency of the Pten KO mice fed with LD or FOD (*n* = 10, 10, respectively). The line within a box indicates the median value. The round dots and cross marks indicate the value of each sample and the mean value, respectively. (**B**) Feed efficiency, dried fecal weight, and digestion efficiency of TRAMP-C2 allograft mice fed with LD or FOD (*n* = 15, 14, respectively). The line within a box indicates the median value. The round dots and cross marks indicate the value of each sample and the mean value, respectively. (**C**) Comparison of alpha-diversity in gut microbiome according to the difference in experimental diets (*n* = 10, 10, respectively). The line within a box indicates the median value, and black dots indicate the value of each sample. (**D**) Comparison of alpha-diversity in gut microbiome according to the types of mouse models (*n* = 10, 10, respectively). The line within a box indicates the median value, and black dots indicate the value of each sample. (**E**) Comparison of beta-diversity in gut microbiome according to the difference in experimental diets. (**F**) Comparison of beta-diversity in gut microbiome according to the types of mouse models. LD, lard diet; FOD, fish oil diet.

**Figure 3 ijms-23-02214-f003:**
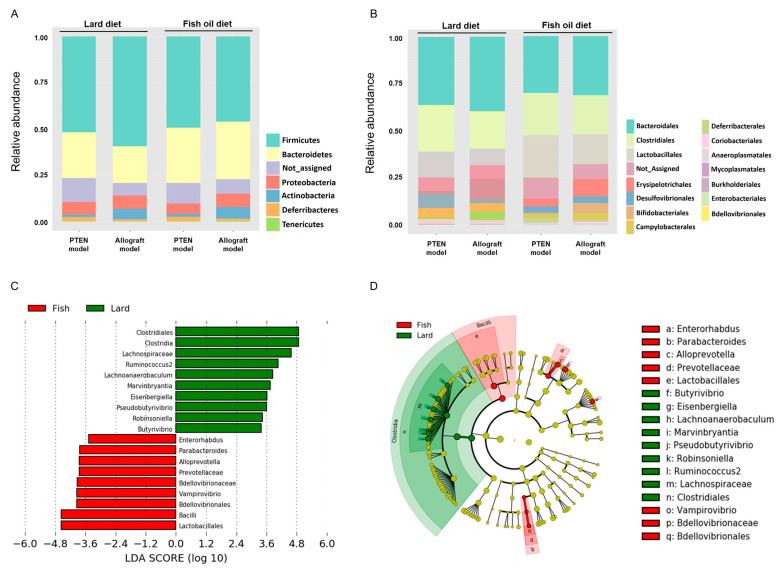
Taxonomic differences in gut microbiota of two mouse models fed with LD or FOD. (**A**,**B**) Relative abundance of microbial groups in fecal samples in the Pten KO and TRAMP-C2 allograft models at the phylum (**A**) and the order (**B**). (**C**) Histograms of linear discriminant analysis (LDA) effect size (LEfSe) of gut microbiota between the LD and FOD groups. Log-level changes in LDA score are displayed on the X axis. Green bars: taxa found in greater relative abundance in the LD group. Red bars: taxa found in greater relative abundance in the FOD group. (*p* < 0.05 and LDA score (log10) > |3|). (**D**) Taxonomic cladogram using the LEfSe method indicating the phylogenetic distribution of gut microbiota associated with LD and FOD. The color of the dots and sectors indicate the compartment in which the respective taxa are most abundant. Green dots and sectors indicate abundant taxa in the LD group, and red dots and sectors indicate abundant taxa in the FOD group. LD, lard diet; FOD, fish oil diet.

**Figure 4 ijms-23-02214-f004:**
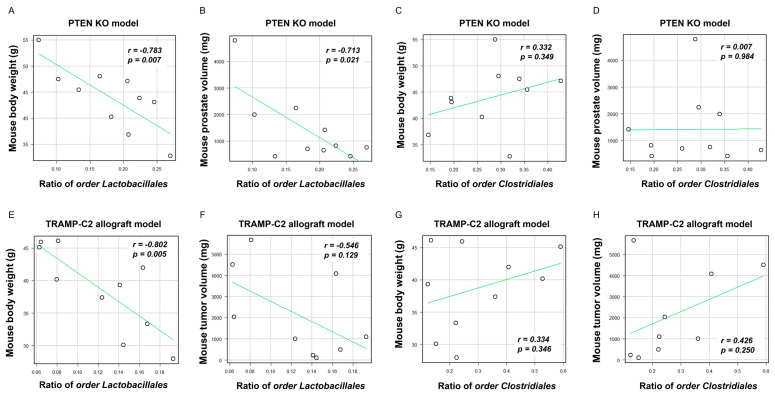
Correlation between the proportion of two specific microbiotas, body weight, prostate weight, and tumor volume. The dots in each scatter plot indicate the values of each sample. (**A**) Correlation between the body weight of mice and the rate of the order Lactobacillales in the Pten KO model. (**B**) Correlation between the prostate volume of mice and the rate of the order Lactobacillales in the Pten KO model. (**C**) Correlation between the body weight of mice and the rate of the order Clostridiales in the Pten model. (**D**) Correlation between the prostate volume of mice and the rate of the order Clostridiales in the Pten KO model. (**E**) Correlation between the body weight of mice and the rate of the order Lactobacillales in the TRAMP-C2 allograft model. (**F**) Correlation between the subcutaneous tumor of mice and the rate of the order Lactobacillales in the TRAMP-C2 allograft model. (**G**) Correlation between the body weight of mice and the rate of the order Clostridiales in the TRAMP-C2 allograft model. (**H**) Correlation between the subcutaneous tumor of mice and the rate of the order Clostridiales in the TRAMP-C2 allograft model.

**Figure 5 ijms-23-02214-f005:**
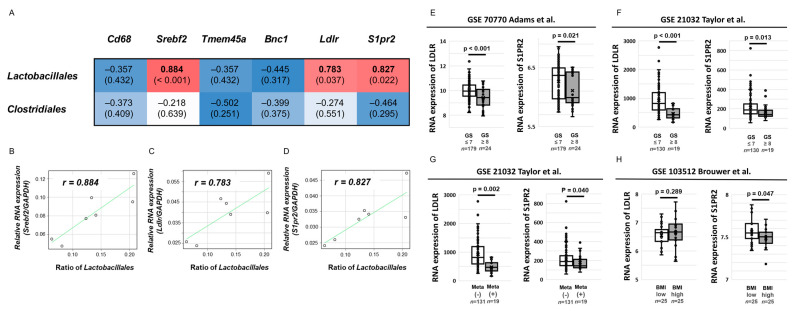
Correlation between the proportion of two gut microbial species and candidate genes that were up- and downregulated in the LD mice and the effect of candidate genes on clinical outcomes in patients with prostate cancer. (**A**) Heatmap for Spearman’s rank correlation coefficient between two gut microbial species, including the orders Lactobacillales and Clostridiales and six candidate genes. Blue color represents a high inverse correlation and red color represents a high positive correlation. (**B**–**D**) Correlation between the proportion of the order Lactobacillales and mRNA expression of *SREBF2* (**B**), *LDLR* (**C**), and *S1PR2* (**D**). The dots in each scatter plot indicate the values of each sample. (**E**–**H**) The effect of *S1PR2* and *LDLR* mRNA expression on clinical outcomes using three Gene Expression Omnibus datasets (GSE 70770, GSE21032, and GSE103512). Differential expression of *S1PR2* and *LDLR* mRNA was statistically assessed according to the Gleason score (**E**,**F**), presence of metastasis (**G**), and body mass index (**H**). The line within a box indicates the median value. The round dots and cross marks indicate the value of each sample and the mean value, respectively.

**Table 1 ijms-23-02214-t001:** Commonly regulated genes in two mouse models of prostate cancer fed with a lard diet and fish oil diet.

Gene Symbol	Gene Description	PTEN KO Model		TRAMP-C2 Allograft Model	
		Abundance Ratio(Lard/Fish)	*p*-Value	Abundance Ratio (Lard/Fish)	*p*-Value
2010005H15Rik	RIKEN cDNA 2010005H15 gene	1.468	0.05	1.399	0.008
Gm22043	predicted gene, 22043	1.351	0.012	1.252	0.02
D130007C19Rik	RIKEN cDNA D130007C19 gene	1.309	0.016	1.323	0.019
n-R5s220	nuclear encoded rRNA 5S 220	0.736	0.007	0.711	0.029
Cd68	CD68 antigen	0.679	0.042	0.779	<0.001
Srebf2	sterol regulatory element binding factor 2	0.754	0.002	0.655	0.022
Tmem45a	transmembrane protein 45a	0.662	0.025	0.742	0.007
Bnc1	basonuclin 1	0.668	0.048	0.64	0.024
Ldlr	low density lipoprotein receptor	0.661	0.014	0.604	0.011
S1pr2	sphingosine-1-phosphate receptor 2	0.746	0.005	0.759	0.026

**Table 2 ijms-23-02214-t002:** Pathway enrichment analysis in two mouse models of prostate cancer fed with a lard diet and fish oil diet.

Wiki Pathway	Changed Genes	Total Genes	Z Score	*p*-Value
*Cholesterol Bio synthesis*	5	15	9.62	1.27 × 10^−5^
*Cholesterol metabolism (includes both Bloch and Kandutsch-Russell pathways)*	8	48	8.16	2.53 × 10^−6^
*Matrix Metalloproteinases*	5	28	6.72	1.64 × 10^−4^
*SREBF and miR33 in cholesterol and lipid homeostasis*	3	11	6.66	1.32 × 10^−3^
*Endochondral Ossification*	7	62	5.95	1.10 × 10^−4^
*Lung fibrosis*	6	61	5.01	6.86 × 10^−4^
*Prostaglandin Synthesis and Regulation*	4	31	4.91	2.31 × 10^−3^
*Eicosanoid Synthesis*	2	18	3.15	4.10 × 10^−2^
*Small Ligand GPCRs*	2	18	3.15	4.10 × 10^−2^
*Statin Pathway*	2	19	3.03	4.49 × 10^−2^
*Retinol metabolism*	3	39	2.96	3.01 × 10^−2^
*Splicing factor NOVA regulated synaptic proteins*	3	42	2.79	3.60 × 10^−2^
*Adipogenesis genes*	6	133	2.58	2.47 × 10^−2^

**Table 3 ijms-23-02214-t003:** Gene ontology analysis in two mouse models of prostate cancer fed with a lard diet and fish oil diet.

GO Term	Changed Genes	Total Genes	Z Score	*p*-Value
*lipid metabolic process*	15	469	5.97	4.31 × 10^−6^
*aging*	7	165	5.07	3.67 × 10^−4^
*membrane organization*	2	32	3.52	2.82 × 10^−2^
*cell death*	2	35	3.32	3.30 × 10^−2^
*cell adhesion*	10	507	3.06	6.55 × 10^−3^
*embryo development*	3	80	3.03	2.63 × 10^−2^

## Data Availability

Data are contained within the article or Appendix A. Data presented in this study are available in the Appendix A.

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
