# Peer review of "Specific Gut Microbial Environment in Lard Diet-Induced Prostate Cancer Development and Progression"

_ijms, 2022, doi:10.3390/ijms23042214_

Round 1

Reviewer 1 Report

The group convincingly showed how saturated fatty acid diet not only induced the development of prostate cancer, but also modified the diversity of the microbiome. Clear method and results.

As with the discussion, I strongly suggest the authors to discuss the effect of fish oil diet and the reason(s) for the difference in the microbiome diversity between the lard and fish oil diets.

Author Response

January 29, 2022

Editor-in-Chief

Dr. Maurizio Battino

International Journal of Molecular Science

Manuscript ID:  ijms-1491359

Title: Specific Gut Microbial Environment in Lard diet-induced Prostate Cancer Development and Progression

by Sato H. et al.

Dear Dr. Maurizio Battino,

              Thank you very much for your letter of 10th December 2021, with regard to our manuscript together with your reviewer’s comments. The comments of the reviewers have been helpful in allowing us to revise our manuscript. We have revised the manuscript accordingly:

Response to Reviewer 1 Comments

Major point

1, The most serious problem is that the mechanism by which gut microbiota influence gene expression and growth in prostate cancer far from the intestinal tract has not been mentioned. The correlation between the abundance of Lactobacillaes and gene expression does not rule out the potential mechanisms that saturated fatty acid intake may affect the gut microbiota and mRNA expression, independently. In order to prove that saturated fatty acid-influenced gut microbiota is directly involved in gene expression in prostate cancer, additional experiments are needed, in which authors should intervene in the gut microbiota of mice by appropriate methods, such as bacterial transplantation, fecal transplantation, and antibiotics administration.

Response 1: We thank the reviewer for the important comments and completely agree with the reviewer’s opinion. As pointed out, we did not prove in this study how diet-induced gut dysbiosis directly influences gene expression in prostate cancer. However, a previous study using the same GEM prostate cancer model demonstrated that oral administration of an antibiotic mixture in prostate cancer–bearing mice fed a high-fat diet altered the composition of the specific gut microbiota and inhibited prostate cancer cell proliferation as well as the modulation of prostate IGF-1 expression (Matsushita, Cancer Res, 2021). The study also showed that short-chain fatty acids produced by intestinal bacteria were key regulators of high-fat-diet-induced prostate tumor growth. Therefore, the study supports diet-induced prostate cancer progression with the modulation of intraprostatic gene expressions through gut dysbiosis. Moreover, microbiota-derived metabolites including short chain fatty acid, bile acids, and endotoxins are connected to the immune and endocrine systems and associated with disease in the distant organs (Schroeder BO, Nat Med, 2016). These studies partly supported our findings that specific diet-induced gut microbiota alteration accelerates prostate tumor growth and gene expression changes in the prostate through microbiota-derived metabolites. As pointed out by the reviewer, underlying mechanisms associated with lard-diet-induced prostate cancer development found in the present study needs to be clarified. Given that it will take about half of a year to finish and obtain the results of fecal transplantation and/or antibiotic models because of the slow growth of prostate cancer in our mouse models, we have clarified these points in the discussion section with future direction of our study (Ln 382–396).

2, The ingredients of the two high-fat diets are not describes in detail. It would be better to describe the fat compositions (the ratio of saturated fat, etc) and fiber abundance that may affect the gut microbiota composition.

Response 2: We thank the reviewer for the helpful comment. According to the reviewer’s suggestion, we have added details of the two experimental diets including the fiber abundance and the composition of typical fatty acids in the Materials and Methods section and supplementary Tables 8 and 9. Based on the details of the ingredients, our experimental lard diet is enriched not only in saturated but also monounsaturated fatty acids (SMFA). Therefore, we have revised the description of the diets used in this study throughout the manuscript (Ln 12–15, Ln 41–43, Ln 286–288, Ln 315–317, Ln 435–439).

3, In line181-189, it was described that bacteria have correlations between tumor volume and body weight, even though the P-value is under the level of significance or trend (Figre 4C, D, G, and H). Since these correlations may be misleading due to alpha error caused by small samples, the authors should change that there were no significant correlations, or increase the sample size.

Response 3: We agree with the reviewer’s concern about the confusing in Fig. 4C, D, G, H. Following the reviewer’s suggestion, we have revised the sentence in the Results section (Ln 188–189).

4, Genes used in the pathway enrichment analysis and GO analysis is not clear; did you use the 10 genes in Table 1? However, in the methods section it states that 274 genes were imported. Also, did you use both up- and downregulated genes? If yes, it is better to perform the analysis only up- or downregulated genes.

Response 4: We apologize for the confusion regarding the genes used in the microarray analyses. In the pathway enrichment and GO analyses, we imported 274 genes in the two mouse models. We have revised the sentences in the Results section to clarify the details of the experiment (Ln 215–222). We have also conducted additional pathway enrichment and GO analyses using the up- or downregulated genes, separately, following the reviewer’s suggestion. Although the trends of important pathways and signaling were consistent with the results of the previous analysis, the results of separate analysis are described in supplementary Tables 3, 4, 5, and 6.

Minor point 

1, There is no scare bar in Figure 1D, E, and G.

Response 1: We thank the reviewer for the comment. We have modified Fig. 1D, E, and G with a scale bar.

2, (Fig3C, D, G, and H) → (Fig4C, D, G, and H) in line186

Response 2: We thank the reviewer for pointing out the typo. We have corrected it.

3, Gene name in Supplementary Table1 is Nestin-Cre. Is Pb-Cre correct?

Response 3: It is known that the “Nestin-Cre” primer can detect the Cre gene regardless of the type of promoter. Therefore, we used the “Nestin-Cre” primer to identify “Pb-Cre” in the present study.

Reviewer 2 Report

In this manuscript, authors used two prostate cancer mouse models to analyze the effects of different lipid intake on prostate cancer and gut microbiota. Lard diet, enriched saturated fatty acid, induced prostate cancer progression and altered the composition of gut microbiota. Lactobacillales was correlated with mRNA expression of several genes associated with cholesterol synthesis and lipid metabolisms in cancer tissue, indicating that unsaturated fatty acids could promote cancer growth through changes in lipid metabolism by specific intestinal bacteria. The gut microbiota altered by diet affects various diseases other than digestive diseases, which has attracted much attention in recent years. Therefore, this study, which refers the effect of intestinal bacteria on prostate cancer progression, is significant. However, there are serious problems that need to be addressed.

Major points

1, The most serious problem is that the mechanism by which gut microbiota influence gene expression and growth in prostate cancer far from the intestinal tract has not been mentioned. The correlation between the abundance of Lactobacillaes and gene expression does not rule out the potential mechanisms that saturated fatty acid intake may affect the gut microbiota and mRNA expression, independently. In order to prove that saturated fatty acid-influenced gut microbiota is directly involved in gene expression in prostate cancer, additional experiments are needed, in which authors should intervene in the gut microbiota of mice by appropriate methods, such as bacterial transplantation, fecal transplantation, and antibiotics administration.

2, The ingredients of the two high-fat diets are not describes in detail. It would be better to describe the fat compositions (the ratio of saturated fat, etc) and fiber abundance that may affect the gut microbiota composition.

3, In line181-189, it was described that bacteria have correlations between tumor volume and body weight, even though the P-value is under the level of significance or trend (Figre 4C, D, G, and H). Since these correlations may be misleading due to alpha error caused by small samples, the authors should change that there were no significant correlations, or increase the sample size.

4, Genes used in the pathway enrichment analysis and GO analysis is not clear; did you use the 10 genes in Table 1? However, in the methods section it states that 274 genes were imported. Also, did you use both up- and downregulated genes? If yes, it is better to perform the analysis only up- or downregulated genes.

Minor point

1, There is no scare bar in Figure 1D, E, and G.

2, (Fig3C, D, G, and H) → (Fig4C, D, G, and H) in line186

3, Gene name in Supplementary Table1 is Nestin-Cre. Is Pb-Cre correct?

Author Response

January 29, 2022

Editor-in-Chief

Dr. Maurizio Battino

International Journal of Molecular Science

Manuscript ID:  ijms-1491359

Title: Specific Gut Microbial Environment in Lard diet-induced Prostate Cancer Development and Progression

by Sato H. et al.

Dear Dr. Maurizio Battino,

              Thank you very much for your letter of 10th December 2021, with regard to our manuscript together with your reviewer’s comments. The comments of the reviewers have been helpful in allowing us to revise our manuscript. We have revised the manuscript accordingly:

Response to Reviewer 2 Comments

As with the discussion, I strongly suggest the authors to discuss the effect of fish oil diet and the reason(s) for the difference in the microbiome diversity between the lard and fish oil diets.

Response: We thank the reviewer for the valuable comments. As mentioned in our response to reviewer #1, several mechanisms may be expected regarding the underlying mechanism of lard- and fish oil diet-modulated prostate cancer development through the gut microbiota and gene expression changes found in the present study. A previous study using the same GEM prostate cancer model demonstrated that oral administration of an antibiotic mixture in prostate cancer-bearing mice fed a high-fat diet altered the composition of the specific gut microbiota and inhibited prostate cancer cell proliferation as well as the modulation of prostate IGF-1 expression (Matsushita, Cancer Res, 2021). The study also showed that short-chain fatty acids produced by intestinal bacteria were key regulators of high-fat-diet-induced prostate tumor growth. Therefore, the study supports diet-induced prostate cancer progression with the modulation of intraprostatic gene expressions through gut dysbiosis. Although several potential mechanisms need to be considered as mentioned above, further studies are required to clarify our findings. We have added some sentences about this point in the Discussion section (Ln 382–396).

Round 2

Reviewer 2 Report

My comments have been appropriately answered.

I think that the paper has reached the level to be published.